# Gut–Brain Axis: Focus on Sex Differences in Neuroinflammation

**DOI:** 10.3390/ijms25105377

**Published:** 2024-05-15

**Authors:** Mario Caldarelli, Pierluigi Rio, Andrea Marrone, Francesca Ocarino, Monica Chiantore, Marcello Candelli, Antonio Gasbarrini, Giovanni Gambassi, Rossella Cianci

**Affiliations:** 1Department of Translational Medicine and Surgery, Catholic University of Rome, Fondazione Policlinico Universitario A. Gemelli, IRCCS, 00168 Rome, Italy; 2Department of Emergency, Anesthesiological and Reanimation Sciences, Catholic University of Rome, Fondazione Policlinico Universitario A. Gemelli, IRCCS, 00168 Rome, Italy

**Keywords:** gut–brain axis, neuroinflammation, sex differences

## Abstract

In recent years, there has been a growing interest in the concept of the “gut–brain axis”. In addition to well-studied diseases associated with an imbalance in gut microbiota, such as cancer, chronic inflammation, and cardiovascular diseases, research is now exploring the potential role of gut microbial dysbiosis in the onset and development of brain-related diseases. When the function of the intestinal barrier is altered by dysbiosis, the aberrant immune system response interacts with the nervous system, leading to a state of “neuroinflammation”. The gut microbiota–brain axis is mediated by inflammatory and immunological mechanisms, neurotransmitters, and neuroendocrine pathways. This narrative review aims to illustrate the molecular basis of neuroinflammation and elaborate on the concept of the gut–brain axis by virtue of analyzing the various metabolites produced by the gut microbiome and how they might impact the nervous system. Additionally, the current review will highlight how sex influences these molecular mechanisms. In fact, sex hormones impact the brain–gut microbiota axis at different levels, such as the central nervous system, the enteric nervous one, and enteroendocrine cells. A deeper understanding of the gut–brain axis in human health and disease is crucial to guide diagnoses, treatments, and preventive interventions.

## 1. Introduction

The gut microbiota (GM) harbors a huge number of microorganisms, such as bacteria, viruses, protozoa, fungi, and archaea [1]. Overall, 99% of the species is accounted for by Firmicutes, Bacteroidetes, Proteobacteria, Actinobacteria, Fusobacteria, and Verrucomicrobia [2]. These microorganisms can have both beneficial and harmful effects on human health. A healthy balance among the GM is crucial for mental and physical health, as well as for preventing and treating several pathological conditions [3].

There is a growing interest in the brain–gut microbiota axis (BGMA) [4], with research focusing on the possible role of dysbiosis in the onset of brain-related diseases such as multiple sclerosis, Parkinson’s disease (PD) and Alzheimer’s disease (AD), Huntington’s chorea, epilepsy, amyotrophic lateral sclerosis, Guillain-Barré syndrome, and even autism spectrum disorders [5,6].

The central nervous system (CNS), previously considered an “immune privileged” region, has bidirectional cross-talk with the immune system, whereby the immune cells support cerebral function and neuronal repair and neurons influence the immune response by innervating lymphoid organs [7].

Important CNS cell types include neurons, glial cells (including astrocytes, oligodendrocytes, microglia, and ependymal cells), choroid plexus cells, and blood vessels [8]. Tissue-resident macrophages, called “microglia”, act as a baseline immune sentinel [9]. The peripheral immune cells circulate in meningeal spaces and contribute to brain function through the production of cytokines, such as interleukin (IL)-4, IL-17, IL-5, and IL-13, interferon (IFN)-gamma, and acetylcholine (ACh). The meninges also allow the lymphatic drainage of brain-derived soluble molecules, thus playing an important role in neuroimmunology [9].

In physiological conditions, the brain immune system recognizes potential offenders, such as bacteria, viruses, and protein aggregates, and orchestrates an immune response to remove the pathogens and heal brain tissue damage. Damage- and pathogen-associated molecular patterns (DAMPs and PAMPs) stimulate the inflammatory surface receptors of glial cells, activating signal transducers and transcription factors responsible for releasing inflammatory mediators. The immune response is generally self-limited, but, in some cases, there is an exaggerated immune response and the development of chronic inflammation, which further increases toxicity. There are many reasons for the failure of immune response and these might be endogenous (e.g., protein aggregates), environmental (e.g., diet, gut dysbiosis, or infections), or due to genetics (e.g., progranulin or apolipoprotein E4 mutations). In addition, the production of specialized pro-resolving lipid mediators (SPMs), normally involved in the resolution of the inflammatory response, is reduced, and that has also been linked to chronic neuroinflammation [10].

Sex is a biological variable, influencing various aspects of human health and disease [11]. Sex differences have been identified in neuroimmune responses (e.g., glial cell activation and cytokine production), possibly explaining sex-based differences in the onset of neuroinflammation and the subsequent cognitive decline and behavioral disorders [12].

Recent research has elucidated the influence of sex-related factors on the BGMA [13,14,15,16]. The modulation of the immune system by the BGMA and sex hormones is an important area of research [17].

This narrative review aims to illustrate the intricate interplay between the gut microbiome and the nervous system, describing the molecular patterns underlying neuroinflammation and how are they affected by sex differences. 

We have conducted a systematic search of electronic databases including PubMed, MEDLINE, and Google Scholar using keywords such as “neuroinflammation”, “sex-differences”, “gut microbiome”, “gut–brain axis”, and “neurotransmitters”. We considered original and review articles, meta-analyses, and systematic reviews written in English between 2009 and 2024. We selected articles based on study design, methodology, and sample size.

## 2. Neuroinflammation

Neuroinflammation is a critical mechanism in the development of neurological disorders and cognitive aging [18]. Overall, neuroinflammation includes four different mechanisms: an increased brain concentration of pro-inflammatory molecules (e.g., reactive oxygen species (ROS), prostaglandins, cytokines, chemokines, and matrix metalloproteases), the functional activation of microglia and astrocytes in specific brain areas, the infiltration of peripheral immune cells (e.g., macrophages and T-lymphocytes cells) due to a “leaky” blood–brain barrier (BBB), and, finally, the death of neuronal cells [19].

Neuroinflammation can be triggered by peripheral inflammation (e.g., gastrointestinal inflammatory disorders, rheumatoid arthritis, chronic obstructive pulmonary disease, and dermatitis) through various mechanisms. For example, an impaired intestinal barrier, or “leaky gut”, such as in inflammatory bowel diseases (IBD) or celiac disease, can provoke a systemic spread of microbial metabolites and pathogens. These can compromise BBB integrity and, consequently, induce neuroinflammation [20]. In AD, there is some evidence that a GM-mediated release of cytokines facilitates the formation of amyloid-β (Aβ) plaques and neurofibrillary tangles (NFT) [21]. In the case of systemic inflammation, pro-inflammatory cytokines enter the CNS directly or by disrupting the BBB. However, in such conditions, glial cells themselves release inflammatory mediators, activate pathways leading to neuronal death, and promote the production of neurotoxic factors by astrocytes [22]. Inflammatory astrocytes are responsible for a dysfunctional exchange between cerebrospinal and interstitial fluid (glymphatic clearance), which in normal conditions allows the clearance of toxic substances from the CNS [23,24]. A different distribution of neurons and glial cells in the brain could explain regionally localized neuroinflammation [22]. As detected by Do and Woo in a dextran sulfate sodium salt (DSS)-induced colitis mouse model, IBDs can cause brain inflammation to a different extent in specific CNS areas. The authors observed a hippocampal upregulation of cyclooxygenase-2 (COX-2) and glial fibrillary acidic protein (GFAP) during exposure to DSS, a hypothalamic upregulation of COX-2 only a few days after DSS exposure, and a downregulation of COX-2 and brain-derived neurotrophic factor (BDNF) in the amygdala [25].

Neuroinflammation can also be caused by occupational injuries or toxicants, such as industrial chemicals or heavy metals. Exposure to these substances induces “reactive gliosis”, which by activation of microglia and astrocytes culminates in a neuroinflammatory response, neurotoxicity, and neurodegeneration, as observed in traumatic brain injury [26]. 

Physical exercise regulates many neurophysiological aspects, including autophagy, neuronal plasticity, and antioxidant and anti-inflammatory responses, preserving brain function and attenuating neurodegeneration [27]. Long-term low- or moderate-intensity physical exercise reduces the inflammatory response [28]. Aerobic exercise increases mRNA expression of the ATP-binding cassette transporter A1 (ABCA1), potentially enhancing cognitive function and ameliorating symptoms in AD patients [29]. Moreover, exercise may reduce and delay the onset of severe neuropsychiatric symptoms such as apathy, confusion, and depression [30]. High-intensity physical exercise increases levels of BDNF, a neurotrophin expressed in the hippocampus and involved in various neurophysiological processes, such as memory, learning, inflammatory response, and neuronal survival [27,31].

Diet affects neuroinflammation through various mechanisms. Short-chain fatty acids (SCFA) deriving from GM fermentation of indigestible foods are able to cross the BBB and impact the function of nearby cells [32]. Astrocyte activity is influenced by metabolites of dietary tryptophan, such as indole-3-aldehyde and indole-3-propionic acid. These metabolites bind the aryl hydrocarbon receptors (Ahr) present on astrocytes and through transforming growth factor α (TGF-α) and vascular endothelial growth factor B (VEGF-B) serve as pro-inflammatory mediators [33]. TMAO (trimethylamine-N-oxide) and secondary bile acids, produced by microbiota, influence metabolism and neuroinflammation through Farsenoid X (FXR), Takeda G-protein-coupled receptor 5 (TGR5), and glucocorticoid receptor [34,35]. High salt consumption hyperactivates T helper 17 (Th17) cells and the IL-17-mediated cascade of endothelial nitric oxide (NO) through endothelial NO synthase (eNOS), which causes changes in vascular permeability and cerebral circulation of immune cells [36].

Furthermore, the metabolites produced by ketone bodies derived from a ketogenic diet can modulate neuroinflammation mechanisms through mitochondrial function and oxidative stress [37]. Beta-hydroxybutyric acid (βHB), the main ketone product together with acetoacetate (AcAc), activates the hydrocarboxylic acid receptor 2 (HCA2), expressed in microglia, producing prostaglandin D2 (PGD2), which has a positive effect on neuroinflammation [38]. Moreover, βHB reduces the inflammasome activation process, resulting in a reduction of IL-1β [39]. Another mechanism is the anti-inflammatory activity mediated by the activation of peroxisome proliferator-activated receptors (PPARs), transcription factors involved in glucose and lipid metabolism [40]. This suggests that the ketogenic diet may be beneficial in neurodegenerative diseases. In AD, it has been associated with a better clinical outcome and a reduction of amyloid accumulation and microglial activation, and similar findings have been shown in patients with multiple sclerosis [41,42].

A high-fat diet (HFD), on the other hand, contributes to neuroinflammation and cognitive impairment. HFD-fed mice show higher inflammatory parameters and oxidative stress and lower mitochondrial oxidative capacity than standard diet–fed mice. This is linked to a reduced expression in HFD of BDNF, which is involved in neuronal plasticity and energy metabolism [43]. The release of IL-1β, one of the most important mediators of neuroinflammation, depends on the activation of the P2X7 receptor (P2X7R)-inflammasome complex [44]. The stimulation of P2X7R, preferentially localized on microglia, increases the production of neurodegeneration-inducing molecules such as cytokines, chemokines, reactive oxygen and nitrogen species, and proteases [45]. Interestingly, in a novel study on mice, Rossi et al. reported that the lack of P2X7R is protective against the neuronal damage induced in crucial cognitive areas by HFD exposure [44]. 

The effects of dietary metabolites on CNS are summarized in Table 1.

At a molecular level, neuroinflammation and neurodegeneration often result from the activation of inflammasomes, intracellular sensors expressed by several CNS-resident cells, especially microglia but also astrocytes and neurons, and periphery-derived myeloid cells [46]. These cells recognize pathogenic agents through pattern recognition receptors (PRRs), which can be membrane-bound (e.g., Toll-like receptors–TLRs) or intracellular (e.g., nucleotide-binding domain and leucine-rich repeat-containing receptors–NLRs, and AIM2-like receptors–ALRs). The cytosolic oligomerization of PRRs forms multiprotein complexes, namely inflammasomes, responsible for the recruitment of procaspase-1 and its activation in caspase-1. In turn, this activation leads to the maturation of pro-IL-1β and pro-IL-18 into active inflammatory cytokines [46]. For this reason, once activated, inflammasomes promote an innate immune response against harmful agents through IL-1β and IL-18 but also induce pyroptosis, a form of cell death that provides additional inflammatory stimuli [47]. In the context of neurodegenerative diseases, the most investigated inflammasome is the nucleotide-binding domain (NOD), leucine-rich repeat (LRR), and pyrin domain-containing protein-3 (NLRP3) inflammasome.

Intense crosstalk between TLRs and the inflammasome pathway has been described in neurodegenerative diseases since the TLR activation acts as a priming signal for the expression of NLRP3, pro-IL-18, and pro-IL-1β. Among the TLRs, the main actors in neurodegenerative diseases are TLR2, TLR4, and TLR9. Once activated, TLR2 promotes the myeloid differentiation primary-response protein 88 (MyD88)-dependent pathway, leading to the activation of mitogen-activated protein kinase (MAPK) and nuclear factor kappa B (NF-kB) and the production of pro-inflammatory cytokines. TLR4 also induces a MyD88-independent pathway involving the IFN regulatory factor 3 (IRF3), which leads to the release of type I IFNs. Furthermore, TLR9 is an important sensor located on intracellular vesicles, which binds CpG (cytosine nucleotide followed by guanine nucleotide) dinucleotides and activates a MyD88-dependent signaling with the nuclear translocation of IFN regulatory factor 7 (IRF7) and the production of type I IFNs [10].

Examples of molecular mechanisms of neuroinflammation and neuronal death in AD are apoptosis, necroptosis, and NLRP3-mediated neuroinflammation, triggered by the accumulation of Aβ and NFT in the brain. These mechanisms share signaling pathways. The activation of tumor necrosis factor/nerve growth factor (TNF/NGF) receptors recruits caspase-8, forming the death-initiating signaling complex (DISC), which leads to apoptosis. However, when caspase-8 is inhibited, receptor-interacting protein (RIP) kinases 1 and 3 (RIPK1 and RIPK3) form the necrosome and initiate necroptosis [48]. A necrosome-mediated activation of NLRP3 inflammasome, which is negatively regulated by caspase-8, takes part in the molecular pathogenesis of AD [49,50]. Moreover, other types of neuronal death, such as ferroptosis, pyroptosis, and PANoptosis, seem to contribute to the brain damage observed in AD [48].

It is noteworthy that a recent study by Srinivasan and colleagues questioned the role of inflammasome in AD. The authors evaluated the inflammasome signaling in both microglia and the whole body in mice with Aβ-induced AD and found that the deletion of some inflammasome effectors, such as caspases 1 and 11, did not significantly impact amyloid pathology or disease progression [51].

Several studies explored the role of NLRP3 inflammasome in Parkinson’s disease (PD). Higher expression of NLRP3 and caspase-1 has been observed in the substantia nigra of a mouse model of PD [52]. Additionally, the inhibition of the NLRP3/caspase-1/IL-β axis has been shown to protect dopaminergic neurons from damage and improve motility in PD models [53]. In the PD brain, α-synuclein acts as a DAMP, entering cells via TLR-2 and causing a pro-inflammatory shift [54]. Moreover, adaptive immunity contributes to neuroinflammation in PD. For instance, CD4^+^ and CD8^+^ T cells infiltrate the CNS of patients with PD, especially CD8 T cells in the early stage of disease [55]. Additionally, genetic deletion of T cell receptor beta (TCRβ) or CD4 has proved to reduce the major histocompatibility complex II (MHCII) response of CNS myeloid cells to α-synuclein, as well as the dopaminergic cell loss caused by α-synuclein overexpression [56].

NLRP3 inflammasome-mediated neuroinflammation is closely related to the pathogenesis of some mental disorders, such as depression. Two signaling pathways involved in the NLRP3 inflammasome activation have been described in depression: the priming process and the protein complex assembly one [57]. In the priming process, glucocorticoids, often increased in depressed individuals, bind to their corresponding receptor (glucocorticoid receptor, GR), thus promoting the NF-κB pathway and the production of NLRP3, pro-IL18, pro-IL1β, IL6, and TNF-α. On the other side, in the protein complex assembly process, the activation of GR induces the production of ROS. ROS, together with autophagy-lysosomal pathway dysfunction and potassium (K^+^) efflux, occurring when extracellular adenosine triphosphate (ATP) binds to the microglial purinergic receptor P2X7R, activates the NLRP3 inflammasome [57].

Neuroinflammation has been associated with central sensitization and the development of chronic migraine (CM) [58]. Recent research on migraine has highlighted the role of tryptophan, the related serotonin (5-HT) and kynurenine pathways, and their catabolites. In particular, the role of 5-HT in trigeminal pain processing has been recognized. Moreover, the vasoconstriction effects produced by 5-HT, whose receptors are found in the trigeminal nerve and cranial vessels, have been considered. Those receptors’ agonists are deemed to relieve the intensity of migraine attacks [59]. Scholarly attention has also focused on the action of kynurenic acid, which appears to reduce neuropeptides associated with migraine. An interesting avenue of therapeutic research could be the development of synthetic analogs of kynurenic acid with an enhanced ability to pass the BBB [59].

Neuroinflammatory injury can be exacerbated by a trigger receptor 1 expressed on myeloid cells (TREM1), which is located on granulocytes and monocytes, as well as on microglia [60], and regulates the NF-κB pathway [61]. As observed in a nitroglycerin-induced CM model, TREM1 is upregulated in microglia of the trigeminal nucleus caudalis, where it exerts a pivotal role in the NLRP3 inflammasome activation via NF-κB signaling [62]. 

A chronic neuroinflammatory state with sustained microglia activation is also present in processes such as aging, immunosenescence, mitochondrial dysfunction, obesity-induced meta-inflammation, and gut dysbiosis [63,64]. 

Over the past few years, the role of non-coding RNAs (ncRNAs) in modulating microglia- and astrocyte-mediated neuroinflammation has been explored. Regulatory ncRNAs, which can be distinguished into microRNAs (miRNAs), circular RNAs (circRNAs), and long non-coding RNAs (lncRNAs) [65], may influence the glial cell gene expression in a pro-inflammatory way [66].

The main causes and pathophysiological mechanisms of neuroinflammation are summarized in Table 2.

## 3. The Gut–Brain Axis

The gut–brain axis represents a bidirectionally interconnected system, involving immune, endocrine, and neuronal elements [67]. 

A growing number of different gut microbial species are now believed to modulate brain function in healthy conditions and disease states. The gut microbiota comprises approximately 10^14^ microbial cells, primarily dominated by two phyla, Bacteroidetes and Firmicutes, while others, such as Proteobacteria, Actinobacteria, Fusobacteria, and Verrucomicrobia, represent a smaller proportion but interact with various intestinal microorganisms [68]. Several factors, including age, geographical location, diet, medications, exposure to toxins, infectious agents, and host genetics, can influence the composition and functions of a healthy microbiota [68].

The gut microbiota can produce neurotransmitters, such as 5-HT, dopamine, and γ-aminobutyric acid (GABA).

Over 90% of the body’s 5-HT is synthesized in the gut. Several bacteria, including *Streptococcus* spp., *Enterococcus* spp., *Escherichia* spp., *Lactobacillus plantarum*, *Klebsiella pneumoniae*, and *Morganella morganii*, can produce 5-HT [69]. 5-HT is synthesized from the essential amino acid L-tryptophan (Trp), which is mainly obtained from dietary sources [70]. Enterochromaffin cells synthesize 5-HT in the presence of certain cofactors, including vitamins B6 and B3 and magnesium [71]. This synthesis occurs from its precursor, L-tryptophan, in a reaction catalyzed by the enzyme tryptophan hydroxylase [71].

The gut microbiota promotes the production of enteric 5-HT through SCFAs, as well as phenolic and indolic compounds derived from microbes [71]. SCFAs, such as acetate, propionate, and butyrate, are carboxylic acids derived from the microbial fermentation of complex polysaccharides that are indigestible by the host [72]. In humans, the highest levels of gastrointestinal SCFAs can be found in the colon [73]. SCFAs serve as energy substrates for enterocytes and colonocytes, thereby influencing the integrity and function of the intestinal epithelial barrier [73]. Additionally, SCFAs are involved in anti-inflammatory effects by regulating the recruitment and migration of immune cells, influencing the differentiation of T and B cells, and modulating the gene expression of inflammatory chemokines and cytokines [74]. Butyrate and propionate can reduce the activity of NF-κB and inhibit the secretion of the inflammatory factor TNFα [73]. 

The microbiota also influences the expression of the serotonin transporter (SERT) through gut bacteria via post-translational and transcriptional mechanisms. This includes alterations in SERT surface levels, as well as epigenetic or immune mechanisms [71].

Dopamine, also known as 3,4-dihydroxyphenethylamine, is a primary catecholaminergic neurotransmitter that plays a significant role in various brain functions [75]. Dopamine is synthesized through the phenylalanine–tyrosine–dopa–dopamine pathway. In this way, L-phenylalanine is converted to L-tyrosine by phenylalanine hydroxylase, primarily occurring in the liver and kidneys. L-tyrosine, obtained from the diet or synthesized in the liver and kidneys, can cross the BBB and enter the brain. Within the brain, it is further converted to (S)-3,4-dihydroxyphenylalanine (L-dopa) by the enzyme tyrosine hydroxylase. Finally, L-dopa is transformed into dopamine by dopa decarboxylase [76]. Tyrosine hydroxylase, the rate-limiting enzyme in this process, functions as a monooxygenase and requires tetrahydrobiopterin (BH4) as a cofactor for its enzymatic activity [77]. The gut microbiome can produce BH4. Additionally, it has been observed that gut microorganisms have metabolic pathways such as the phenylalanine-tyrosine-dopa-dopamine found in humans [76]. Indeed, several bacteria have been identified as capable of producing dopamine within the gut, such as *Escherichia coli*, *Proteus vulgaris*, *Serratia marcescens*, *Staphylococcus aureus*, *Hafnia alvei*, and *Klebsiella pneumoniae* [78]. While it is known that certain bacteria in the gut can produce dopamine, the exact mechanism has not yet been fully elucidated [76].

GABA is a non-protein amino acid found ubiquitously in most life forms, and it is considered the major inhibitory neurotransmitter in the brain [79]. Additionally, GABA acts as a “postbiotic”, defined as “a preparation of inanimate microorganisms and/or their components that confers a health benefit on the host”, according to the definition of the International Scientific Association of Probiotics and Prebiotics [80]. Several gut bacteria have genes encoding for the enzyme glutamate decarboxylase (GAD), which catalyzes the synthesis of GABA by utilizing glutamate, CO_2_, and a proton in the presence of the cofactor pyridoxal 5′-phosphate (PLP) [81]. The GAD enzyme is found in bacteria in both pathogenic strains, such as *Escherichia coli* and *Listeria monocytogenes*, as well as in non-pathogenic bacteria, including *Lactococcus lactis*, *Lactobacillus reuteri*, and species from the genera *Bifidobacterium* and *Bacteroides* [82]. Konstanti et al. demonstrated the ability of *Akkermansia muciniphila* to produce GABA in response to low pH when the GABA precursors glutamate or glutamine are present in the medium. It is plausible that *Akkermansia muciniphila* can mitigate the effects of low pH in the medium through the production of GABA [81]. Kaur et al. found that *Lactobacillus fermentum* L18, which produces GABA, has the potential to enhance the gut barrier by upregulating the production of junction proteins. L18 also influences the composition of fecal microbiota by increasing the abundance of beneficial microorganisms [83].

Indeed, the term “psychobiotics” was coined to describe probiotics or prebiotics that have the potential to positively influence mental health by modulating the gut microbiota, as summarized in Table 3. 

Extracellular vesicles (EVs) derived from psychobiotic bacteria are small enough to be absorbed from the gastrointestinal tract and transported to the brain. Once in the brain, these EVs can influence various brain processes [84]. This interaction may occur through various mechanisms, including modulation of the expression of neurotrophic factors [85], regulation of neurotransmitters [86], or potential supplementation of astrocytes with glycolytic enzymes [87].

Several studies have suggested that an imbalance in gut bacteria could potentially contribute to a decrease in monoamines and play a role in the pathophysiology of depression [88].Certain models have suggested that LPS may be linked to major depressive disorder (MDD) [89]. It has been documented that LPS can activate microglial cells and immune cells in the CNS, triggering inflammatory reactions that lead to the death of dopaminergic neurons [90]. Moreover, SCFAs influence neural responses through several pathways. These include the stimulation of the maturation and homeostasis of microglial cells and the inhibition of histone deacetylase activity, which in turn alters gene expression [91].

Additionally, SCFAs trigger the release of intestinal neuropeptides, such as YY peptide (YYP) and GLP-2, along with hormones crucial for maintaining the integrity of the intestinal barrier and cellular metabolism [92].

In patients experiencing depressive symptoms, acetate levels in stool showed a positive correlation with symptoms, whereas levels of butyrate and propionates exhibited a negative correlation [93].

These findings imply that a balance between specific levels of SCFAs and the microbiota is crucial to maintain mental well-being.

Furthermore, inflammation triggered by dysbiosis plays a role in disrupting tryptophan metabolism. IFN-γ and TNF stimulate the enzyme Indoleamine 2,3-dioxygenase (IDO), which favors the production of kynurenine over 5-HT, thereby increasing the risk of MDD [94]. Cytokines released by immune cells residing in the large intestine can trigger the activation of the hypothalamic–pituitary–adrenal (HPA) axis. This activation leads to brain stimulation via cortisol, which further activates the immune response. Inflammatory signals are then transmitted through different cellular, humoral, and neural pathways, activating resident immune cells in the brain. This disturbance in neuronal integrity occurs through alterations in neurotransmitter production [95]. Similarly, concerning anxiety spectrum disorders, several studies have revealed a significant decrease in both the abundance and diversity of microbial populations in individuals diagnosed with generalized anxiety disorder (GAD) [96].

Indeed, a reduced number of bacteria SCFA-producers and an increased number of *Escherichia-Shigella*, *Fusobacterium*, and *Ruminococcus gnavu* have been documented in individuals with GAD [97].

### 3.1. The Role of the Immune System

The immune system plays a crucial role in the bidirectional communication between gut microbiota and brain.

Innate immunity within the intestinal lining is characterized by both intra- and extracellular layers composed of various components. Epithelial cells form a physical barrier, while mucins provide a protective mucous layer. Antimicrobial peptides (AMPs) are also present, functioning to prevent pathogens entry by directly killing or inhibiting the growth of microorganisms [98]. Moreover, approximately 70–80% of immune cells in the body reside in the mesenteric lymph nodes. The main components include dendritic cells, macrophages, neutrophils, natural killers, and mast cells [99].

Adaptive immunity serves as a secondary defense mechanism characterized by its specific response to pathogens. One example of this is the production of secretory immunoglobulin A (sIgA) antibodies that can bind to pathogens and prevent adhesion to intestinal epithelial cells [100]. B and T cells play significant roles in regulating the adaptive immune system’s response. Molloy’s study emphasized the critical role of T regulatory cells, particularly FoxP3^+^ regulatory T cells (Tregs), in establishing tolerance toward gut microbes [101].

Immune regulation is governed by the BBB, along with the activity of microglia, astrocytes, and oligodendrocytes. 

Microglia play a crucial role in maintaining brain homeostasis and can be activated by various microbial and immune pathways [102]. Schafer documented an abnormal microglia development in germ-free mice that lacked colonization with specific bacteria, including *Bacteroides distasonis* and *Lactobacillus salivarius* [103]. SCFAs can also stimulate microglial activation via G protein-coupled receptors (GPCRs) [98].

Astrocytes, as crucial support cells in the brain, play a role in immune regulation and can respond to signals from the microbiota. They carry hydrocarbon receptors (AHRs) capable of binding to metabolites produced by the gut microbiota. This interaction can elicit an anti-inflammatory response, helping to maintain brain homeostasis [102].

Tryptophan can activate both microglia and astrocytes. Its conversion to serotonin in the gut and subsequent interaction with the serotonergic system also plays a role in modulating immune responses [98].

### 3.2. The Role of Aging

The GM is increasingly acknowledged as a significant controller of the immune system and cognitive health. Boeheme et al. demonstrated significant changes in the microbiota associated with declining health and frailty among the elderly [104]. They performed fecal microbiota transplantation from either young (3–4 months old) or old (19–20 months old) donor mice into old recipient mice (19–20 months old). Transplanting microbiota from young donors reversed age-related disparities in peripheral and brain immunity, along with alterations in the hippocampal metabolome and transcriptome of aging recipient mice. The microbiota derived from young donors alleviated specific age-related deficits in cognitive behavior when transplanted into aged hosts [104]. 

Aging triggers microglial activation through the activation of the cyclic GMP–AMP synthase (cGAS)–stimulator of interferon genes (STING) signaling pathway [105]. Misfolded proteins and protein aggregates induce microglial activation by disrupting microglial autophagy [106]. Stage-1 disease-associated microglia (DAM) represents a transitional and functional subtype with an increased capacity for phagocytosis initiated through a mechanism TREM2-independent, while stage-2 DAM represents a dysfunctional state initiated by a TREM2-dependent mechanism [107]. The microglia transition into stage-2 DAM is promoted by the spleen tyrosine kinase (SYK) signaling pathway [108]. Dysfunctional microglial-T-cell signaling contributes to neurodegeneration by releasing neurotoxic factors. Microglial activation initiates a self-perpetuating cycle that exacerbates neurodegeneration, as activated microglia promote the spread of protein aggregates to unaffected brain regions [109].

### 3.3. The Role of Environmental Factors

Low-fiber diets, aging, and sleep deprivation all contribute to dysbiosis and loss of integrity of the intestinal barrier. This occurs through a reduction in species that produce SCFAs and degrade fiber, alongside an increase in species degrading mucin. Low-fiber diets can lead to immune depression at both the mucosal and systemic levels by compromising the metabolic action of CD4 T cells, resulting in a state of neuroinflammation [109]. The depletion of mucin allows for direct interaction between microbes and intestinal epithelial cells, triggering the degradation of tight junction proteins (TJPs) [110]. The reduced levels of both mucin and TJPs lead to the disruption of the gut barrier and the development of a “leaky” gut [111]. Fecal microbiota transplant (FMT) studies have shown that the aged gut microbiota decreases the expression of mucin and TJPs [112].

In older mice, beneficial metabolites, particularly SCFAs such as butyrate, are significantly reduced in the gut, specifically in fecal samples [113]. Mishra et al. reported the importance of butyric acid in the gut–brain axis, finding that the decrease in free fatty acid receptor 2 and 3 (FFAR2/3) expression in the older gut is primarily caused by a deficiency in butyrate [112]. Additionally, the study demonstrates that deficiencies in FFAR2 and FFAR3 in the gut worsen brain inflammation, impair cognitive function, and lead to depressive and anxious behaviors in relatively young (7-month-old) mice. These findings suggest that abnormalities induced by GM accelerate age-related behavioral deficits [112], highlighting the importance of butyrate-FFAR2/3 agonism as a potential strategy to counteract the harmful effects of aged gut microbiota on brain health in older adults.

## 4. Neuroinflammation and Sex Differences

It is known that sex differences may influence the mechanisms of neuroinflammation. For instance, sex hormones, mainly estrogen E2 (17β-estradiol) and testosterone, play a neuroprotective role [114] through different modalities, such as neurogenesis, immune response, and the regulation of microglia function and excitotoxicity [115,116]. These mechanisms take part in several disorders, such as ischemic brain damage and some neurodegenerative diseases (e.g., PD and AD), which are caused by both genetic and environmental factors, as multifactorial conditions.

One of the most interesting areas where sex differences determine different clinical outcomes is ischemic stroke, which is prevalent in women at an advanced age. The mechanisms through which this occurs represent the starting point for a different management of the disease and the development of therapeutic interventions [117]. 

In this context, sex hormones can influence the integrity of the BBB, determining different immune responses following the acute ischemic event [118]. Estrogen modifies the BBB by increasing the production of NO through the classic genomic pathway, the non-classic genomic pathway, and the non-genomic pathway. In the classic genomic pathway, the complex formed by estradiol and estrogen receptors (ERs) binds to estrogen response elements (EREs), thus regulating the transcription of eNOS. In the non-classic genomic pathway, estradiol binds to ERs, initiating an intracellular signaling cascade involving several kinases (e.g., mitogen-activated protein kinase, MAPK) and cyclic adenosine monophosphate (cAMP), promoting the eNOS transcription through co-factors (e.g., NF-κB). In the non-genomic pathway, the NO production is the result of the eNOS phosphorylation induced by the activated protein kinase B (Akt). Other mechanisms through which estrogen alters the BBB are the increased matrix metalloproteinases (MMPs), or reduced tissue inhibitors of metalloproteinases (TIMP), and the inhibition of the RhoA/Rho-kinase-2 (ROCK-2) pathway, involved in BBB integrity. However, in inflammatory conditions, E2 has proved to promote the annexin A1 (ANXA1) phosphorylation, reinforcing the tight junctions and mitigating the inflammatory process [118].

In stroke, sex-specific immune responses have been found in the mechanisms of both innate and adaptive immunity [119].

The neutrophil response, which correlates with the severity of brain damage, appears to be sex-related. Circulating neutrophils are more immature in men; moreover, women have an increased type I IFN pathway activity compared to age-matched men [120]. However, mouse stroke models showed a higher amount of CNS infiltrating neutrophils and greater circulating levels of neutrophil-specific cytokines (e.g., granulocyte colony stimulating factor, G-CSF) in older males compared to age-matched females [121].

Microglial cells and the sex-specific chemotactic signals they produce correlate with a different recruitment of systemic immune cells that regulate neuroinflammation in the injured brain site [122]. Preclinical studies showed the existence of sex differences in activated microglia after ischemic stroke. Females present a predominantly caspase-mediated cell death and higher levels of immunosuppressive M2 phenotype microglia, whereas males present a predominantly Poly (ADP-ribose) polymerase 1 (PARP-1)-mediated cell death, as well as higher NO pathway, PPAR-α expression, and TLR2 signaling [123].

As observed in several preclinical studies, in females, dendritic cells migrating from the periphery into the ischemic brain area exhibit higher levels of TLRs [124], resulting in a better antigen presentation compared to males. The different expression of TLRs between the two sexes is linked to the presence of many TLR genes (e.g., TLR2, TLR3, and TLR7) on the X chromosome [125].

Furthermore, T cells play an important role. Women exhibit greater levels of IL-10, produced by Tregs and T helper 2 (Th2) cells, correlating with immunosuppression and worse recovery after ischemic stroke [126]. This seems to be determined by estrogens, particularly 17β-estradiol, which activates macrophages, increases the circulating levels of anti-inflammatory cytokines, such as IL-10 and TGF-β [126,127], and reduces the pro-inflammatory ones, such as IL-1β, IL-6, and TNFα [128].

Furthermore, neuroinflammation plays a key role in PD, which has heterogeneous and sex-related clinical presentation and response to treatment [129]. Pathogenetic mechanisms of disease seem to differ between the two sexes; for example, a sex-dependent gene expression in the striatum and the substantia nigra has been observed [130]. PD affects mostly older men and seems to present in a more severe form in men at the beginning of the disease [131], thus suggesting a protective role of female hormones, particularly circulating estradiol, on the dopaminergic system. However, women have shown a higher mortality and a faster progression of the disease compared to men [132]. 

In PD, a progressive degeneration of dopaminergic neurons of the substantia nigra occurs due to chronic inflammation triggered by increased levels of alpha-synuclein (α-Syn) supported by microglia and astrocyte reactivity [133,134]. Female neurons appear to be less sensitive than male neurons to the factors determining the development of neuroinflammation, such as oxidative stress, reduced mitochondrial mass, excitotoxicity, and martial metabolism, which is regulated by the estrogens that make women less susceptible to iron accumulation [135]. Estradiol itself promotes the production and metabolism of dopamine [132].

Women with early PD have shown an increased inflammatory activation of monocytes and greater IFN-γ signaling [136]. López-Cerdán and colleagues identified sex-differential transcriptomic patterns in PD related to mitochondrial function, oxidative stress, and neuronal cell death. Gene expression in women was linked to lysosomal and mitochondrial dysfunction, alterations of cytoskeletal proteins, and changes in glutamic metabolism, while in men it was linked mainly to oxidative stress, angiogenesis, and inflammation [137]. Furthermore, differences between the two sexes in the quantity and phenotype of microglia and astrocytes in various brain areas have been detected [138]. A mouse study by Brunialti et al. detected the existence of sex-specific microglial responses to glucocerebrosidase inhibition, realized through conduritol-B-epoxide. The Glucosyl ceramidase beta 1 (GBA1) gene, encoding the enzyme glucocerebrosidase, is often mutated in PD. Interestingly, male microglia developed a more pro-inflammatory phenotype, whereas female microglia showed a reduced ability to promote the nuclear factor erythroid 2–related factor 2 (Nrf2)-dependent detoxification pathway in neuronal cells [139].

AD is the most common neurodegenerative disease, mainly affecting women, with an estimated lifetime risk at age 45 of about 20% for women and 10% for men [140]. From a pathogenetic point of view, neuroinflammation causes progressive neuronal atrophy resulting from the accumulation of amyloid plaques composed of aggregates of Aβ and NFT of hyperphosphorylated Tau protein [141].

The differences between the two sexes can be partly explained by the role that estrogens play, mainly through ERβ, in mitochondrial function at the neuronal level [142]. The post-menopausal loss of estrogen would lead to the impaired mitochondrial function observed in AD [143]. Additionally, estrogens take part in the maintenance of normal levels of Aβ, regulating its clearance and metabolism [144,145], and in the reduction of phosphorylated Tau protein through the glycogen synthase kinase-3 beta (GSK-3β), Wnt, and protein kinase A (PKA) pathways [146]. Estrogens also play an important role in neurogenesis, especially in the dentate gyrus of the hippocampus [147]. 

Neurosteroids, synthesized by neuronal and glial cells or derived from systemic circulation, can modulate CNS functions. In neurodegenerative conditions such as AD, neurosteroidogenesis can be altered [148]. For instance, lower cerebral 17β-estradiol levels were observed in women with AD aged 80 years or older compared to healthy controls [149]. Progesterone, similarly to estrogens, plays a neuroprotective role through gamma-secretase [150] and the insulin-degrading enzyme (IDE) involved in the metabolism of Aβ [151]. Age-related reduction in progesterone levels in women correlates with the risk of AD [152]. Several studies have shown a general progesterone reduction in the brain in AD [153]. Moreover, lower levels of dehydroepiandrosterone (DHEA) and dehydroepiandrosterone sulfate (DHEAS) have been observed in some brain areas, such as the striatum, hippocampus, hypothalamus, amygdala, frontal cortex, and cerebellum, and have been associated with higher levels of Aβ and phosphorylated Tau proteins [154].

Furthermore, the latest evidence suggests that estrogen receptor beta (ERβ) might play a role in triggering migraine headaches; ERβ has shown greater impacts on both the central nervous system and the immune system compared to estrogen receptor alpha (ERα) [63]. Given that ovarian hormones can penetrate the BBB, it is probable that the activity of ERβ contributes to neurogenic inflammation and central sensitization in migraine [64]. Estrogens produce biomolecules that reduce the effectiveness of trigeminal neurotransmission, with subsequent excessive cortical hyperexcitability and cerebral vasodilation. Individuals experiencing migraines exhibit elevated 5-HT levels during the headache phase but reduced levels during the pain-free intervals; estrogen can impact the synthesis of neurotransmitters, such as serotonin [65].

## 5. Brain–Gut Microbiota Axis and Sex Differences

Extensive literature research, particularly of studies utilizing animal models, has unveiled reciprocal and causative connections between the BGMA and sex. 

Gut microbial diversity changes throughout the human lifespan and this phenomenon is recognized to be sex-linked [155]. Sex is an important factor influencing the gut microbiota, as the microbial composition and diversity differ between males and females from birth [156]. A study on preterm infants demonstrated that males exhibited reduced α-diversity (the measurement of the variability of species within a sample) and a higher abundance of Enterobacteriales compared to females. Females, on the other hand, showed higher diversity and a greater abundance of Clostridiales, indicating that sex contributes to the dynamic development of the microbiota [156]. 

Sex differences in microbial alpha diversity evolve throughout life, peaking from adolescence to adulthood, with greater diversity observed in young women compared to young men [157]. These variations in the microbiota may be influenced by sex hormones. 

A research study revealed the existence of a higher prevalence of *Eggerthella* in fecal samples from patients with early ovarian failure, correlated with increased circulating metabolites and TGF-β1 levels. Notably, these alterations were reversible through hormone replacement therapy [158]. 

Ovariectomized estrogen-deficient female mice exhibited β-diversity (measurement of differences in composition between two samples) profiles clustering with males rather than intact females [159]. Sakamuri et al. demonstrated that *Akkermansia muciniphila* exhibited lower abundance in gonadectomized female mice compared to those with intact gonads. Moreover, in the presence of β-estradiol, the growth of *Akkermansia muciniphila* exponentially increased, thereby providing evidence for the identification of a bacterium responsive to female sex hormones [160].

These studies highlight the role of sex hormones in shaping and altering the gut microbiota, contributing to the different colonization between sexes. 

Conversely, the microbiota regulates sex hormones, playing a pivotal role in their metabolism. Enzymes such as sulfatase and β-glucuronidase are involved, as they deconjugate steroids into active forms, which are then reabsorbed in the intestine [161,162]. Circulating estrogen levels are significantly regulated by the microbiota, decreasing as the microbiota is destroyed [161]. The interplay between the microbiome and estrogens, through which gut microbes metabolize estrogens and, in turn, are influenced by the estrogenic metabolites, has been defined as the “estrobolome” [163]. 

Many studies in mice have demonstrated the correlation between the microbiota and sex hormones. In one study, a fecal content transplant from a prepubertal male mouse to a prepubertal female mouse resulted in an increase in testosterone levels in females [164]. In another study, the inoculation of certain strains of Lactobacilli in female mice led to changes in serum levels of estrogen and testosterone [165]. The relationship between testosterone and gut microbiota is intricate; testosterone alters the composition of gut microbiota, and, at the same time, gut microbiota also influences the production of testosterone [166]. Marlke et al. demonstrated that, after microbiota transplantation, there was an increase in circulating testosterone levels. These results highlight the involvement of gut microbiota in the progression of the blood–testis barrier (BTB), spermatogenesis, and testicular steroidogenesis [164]. For example, *Clostridium scindens* and *Ruminococcus gnavus* have been found to synthesize dihydrotestosterone and testosterone within the intestine by converting pregnenolone and hydroxy pregnenolone into androgens [167].

Sex hormones also influence the brain–gut microbiota axis at various levels, such as the central nervous system, the enteric nervous system, and enteroendocrine cells. Scattered along the gastrointestinal epithelium, there are specialized sensory cells known as enteroendocrine cells involved in signals, both direct and indirect, to both the enteric and central nervous systems [168]. Estradiol exerts a direct effect on enteroendocrine L-cells through an ERβ-dependent pathway, increasing glucagon-like peptide 1 (GLP-1) production and secretion [169]. GLP-1 is one of the major components of the BGMA. It operates through the GLP-1 receptor, expressed in different brain regions, where the hormone acts as a neuropeptide involved in the stress response and satiety control [170]. However, evidence suggests that since GLP-1 is rapidly degraded through dipeptidyl peptidase-4 (DPP-4), the main pathway through which it transmits satiety signals to the CNS is the vagal afferent pathway of the enteric nervous system [171]. 

It has been observed that progesterone (P4) induces an increase in GLP-1, too, but this occurs apparently through a non-genomic action and only when administered orally [172]. The complex mechanism by which P4 protects the enteric nervous system (ENS) continues to be a topic of ongoing research. Recent studies highlight the protective role of progesterone in the ENS, but further investigations are necessary to fully understand the long-term effects of progesterone treatment on the ENS [173].

Androgens, notably testosterone, also influence the activity of the gut–brain axis, partly facilitating the conversion to estradiol. Evidence primarily derived from rodent models indicates a protective effect of testosterone on visceral pain. Nonetheless, the limited research conducted in humans and partially conflicting results do not allow definitive conclusions regarding the specific influence of testosterone and its interplay with stress in modulating visceral pain and disorders of gut–brain interaction [174,175].

Sex hormones, particularly estrogens, have receptors on enteric neurons by which they modulate the ENS, the “brain-in-the-gut” that regulates essential gut activities, such as peristalsis, epithelial secretion, and immune signaling. A functional study on isolated ileal segments showed that estrogen receptors ERα, ERβ, and G-protein coupled estrogen receptor 30 (GPR30), located in the myenteric plexus in both female and male mice, contribute to neuronal-mediated contractions in female tissues, while only ERα is involved in this process in male tissues. 

Estrogen also has receptors on intestinal mast cells, which, upon activation, release mediators, such as histamine and proteases, that signal to enteric neurons, thus inducing visceral hypersensitivity or altering intestinal muscle contraction [176]. Several investigations have shown that estrogens can increase the number of mast cell populations and their neuroimmune target secretions [177]. In contrast, androgens seem to exert inhibitory effects on mast cell activity [176].

Estrogens also exert their influence at the vagal level, primarily through the expression of ERα on vagal afferent neurons (VAN), with notably lower expression of ERβ and G protein-coupled estrogen receptor (GPER) [171]. E2 positively regulates ERα expression on VAN, governs the density of axonal projections of vagal afferents in the brainstem [178], and amplifies the excitability induced by mechanostimulation of gastric vagal afferents [171], thus playing a crucial role in VAN functions and potentially contributing to the regulation of food intake. Estrogen receptors are also present throughout the brain, and they may modulate the expression of specific receptors involved in pain processing, such as µ-opioid receptors [179]. This may be related to the increased activation of emotional circuits in women compared to men with irritable bowel syndrome (IBS) [180].

Experimental data indicates that nerve cells exposed to P4 demonstrate increased resilience when subjected to conditions mimicking PD, suggesting the potential role of progesterone receptors within the enteric nervous system in neuroprotection. Additionally, P4 exhibits a range of neuroprotective and neuroplastic effects throughout both the central and peripheral nervous systems [181]. 

Similarly, testosterone has a modulatory effect on neuroinflammation. Malgorzata et al. demonstrated that weak androgen receptor (AR) agonists, which may exhibit beneficial effects in diseases involving neuroinflammation, also possess favorable pharmacokinetics for therapy. They can penetrate the CNS through the blood–brain and blood–cerebrospinal fluid barriers. Given that CXCL1 has been implicated in numerous neuroinflammatory disorders and their respective models, including experimental autoimmune encephalomyelitis (EAE), multiple sclerosis, other demyelinating diseases, neurodegeneration, and infection, modulating its secretion at its sources could potentially offer a common therapeutic approach for seemingly unrelated pathologies [182,183].

Sex hormones also play an important role in the smooth muscle cells of the gastrointestinal tract. Specifically, estrogens relax gastric muscle cells via a nitric oxide- and cyclic guanosine monophosphate-dependent mechanism in a sex-dependent manner, and the relaxation is more pronounced in females than in males [184]. There is limited evidence regarding the effect of androgens on gastrointestinal motility, as conflicting studies exist in the literature. One study demonstrated that androgens induce contraction of intestinal muscle cells in male mice through a non-genomic pathway [185] (Figure 1).

## 6. Future Perspectives

Understanding the molecular basis of neuroinflammation could lead to the development of therapeutic strategies against several neurological diseases.

Diet has proved to exert immunomodulatory and neuroprotective effects and limit the inflammation induced by glial cells, thus improving neurological clinical status [18]. Various studies have explored the neuroprotective role of polyphenolic compounds (e.g., resveratrol, curcumin, and pterostilbene) that can inhibit the NF-kB pathway, as well as prevent the NF-kB nuclear translocation and inflammatory cytokine production [186].

In recent years, the anti-inflammatory effect of some plants, classified as adaptogens (e.g., *Eleutherococcus senticosus*, *Rhodiola rosea*, and *Schisandra chinensis*) has been discovered. They can modulate the initiation and propagation of inflammation, impacting gene expression, signaling pathways (e.g., NF-kB, MAPK, and inflammasome formation) and the production of pro-inflammatory mediators, such as IL-1β, IL-6, INF-γ, prostaglandins, and leukotrienes [187].

Interestingly, Alizadehmoghaddam et al. found that crocin, a natural product derived from saffron and able to cross the BBB, may decrease neuroinflammation in PD by reducing IL-1β and caspase-1 levels and interfering with the gene expression of the NLRP1 inflammasome [188].

The intricate signaling of the NLRP3 inflammasome provides multiple therapeutic options. Some molecules can inhibit different mechanisms, such as the upstream signal transduction, inflammasome assembly, caspase-1 activation, and cytokine release, by interacting with corresponding targets. For instance, among the NLRP3-related inhibitors, CY-09 and Dapansutrile (OLT1177) bind to NLRP3 and inhibit its ATPase activity, while Tranilast and Oridonin bind to NLRP3, respectively inhibiting NLRP3–NLRP3 interaction and the subsequent oligomerization of the adapter protein apoptosis-associated speck-like protein containing a CARD (ASC) and NLRP3-NEK7 (NIMA-related kinase 7) interaction [189]. The administration of MCC950, a small-molecule NLRP3 inhibitor, in mouse PD models reduced the α-synuclein-mediated inflammasome activation, motor deficits, and nigrostriatal dopaminergic degeneration [52].

Kato et al. showed that in a mouse model of high glucose-stimulated microglia, a novel antidiabetic agent, imeglimin, has recently proved to reduce intracellular ROS levels, improve mitochondrial function, and inhibit the activation of the NLRP3 inflammasome, thus suppressing the production of IL-1β [190]. In addition, imeglimin enhances the Unc-51 like autophagy activating kinase 1 (ULK1)-mediated suppression of the thioredoxin-interacting protein (TXNIP)–NLRP3 axis. For this reason, imeglimin could be a promising approach to prevent diabetes-related cognitive impairment [190].

Interestingly, minocycline, an antibiotic with anti-inflammatory properties, was effective in improving cognitive functions and lessening hippocampal damage in a lipopolysaccharide (LPS)-induced neuroinflammatory mouse model. At a molecular level, minocycline reduced the expression of ionized calcium binding adaptor molecule 1 (Iba-1), a marker for microglial activation, as well as the expression of NLRP3/caspase-1, IL-18, and IL-1β [191]. Minocycline, which is also a PARP inhibitor, showed a neuroprotective role in male mice with stroke but not in female ones. Consequently, a potential therapeutic use of minocycline in acute stroke has been suggested; however, sex-related differences are not yet adequately explored [192].

In addition, behavioral strategies, such as physical exercise, have demonstrated a positive impact on neuroinflammation. Li et al. studied a transgenic AD mouse model after several months of physical exercise and detected a significant downregulation of NLRP3, IL-1β, and amyloid-β1-42 expression, a reduction of microglial activation and hippocampal neuronal injury, and enhanced cognitive performance [193]. 

These data support the importance of controlling inflammation to obtain beneficial effects in brain-related diseases. Embracing a sex-oriented perspective in the evaluation of these interventions may increase their efficacy and allow a better interpretation of clinical findings.

## 7. Limitations and Suggestions

The research on neuroinflammation presents several limitations. First, most of the available studies describe the mechanisms and the degree of inflammation in a disease-specific manner, thus lacking a comparative evaluation between different diseases. Moreover, studies generally analyze neuroinflammation in a few brain areas only, failing to provide an overall vision of what happens in the whole CNS [22].

As discussed above, sex plays an influence at several levels the BGMA. However, most studies are based on animal models, and in transitioning from animal to human models, researchers need to be cautious about the unique challenges that human research may present [13].

Moreover, studies on the BGMA have traditionally relied on sex as a dichotomous variable, while the relationship between the BGMA and sex has been little explored. 

Further research is needed to better understand the crosstalk between genetics, epigenetic mechanisms, endocrine pathways, and molecular signaling, collectively contributing to sex differences in neuroinflammation and gut–brain interplay.

## 8. Conclusions

The gastrointestinal tract and the brain are crucial sensors for internal and external signals, and they exhibit an intricate interplay in both physiological and pathological conditions.

This narrative review sheds light on the crosstalk between the gut and the brain, describing the immunological mechanisms, the neurotransmitters, and the neuroendocrine pathways involved in this process. In addition, emerging data support the existence of sex differences at various levels of the BGMA, although their medical implications need to be further clarified.

A deeper understanding of the relationship between sex and the BGMA, involved in human health and disease manifestations, appears crucial to guide diagnoses, treatments, and preventive interventions.

## Figures and Tables

**Figure 1 ijms-25-05377-f001:**
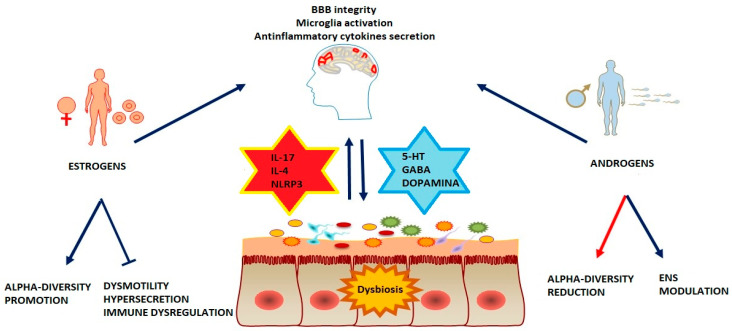
The connection between the gut and brain is bidirectional: an imbalance in gut microbiota results in the decreased production of key neurotransmitters (5-HT, GABA, and dopamine), triggering neuroinflammation, while the nervous system influences dysbiosis and inflammation. Sexual hormones exhibit anti-inflammatory properties in both the central and enteric nervous systems. In particular, estrogens promote a beneficial impact on intestinal microbiota, regulating motility and exocrine function, whereas testosterone leads to a decrease in bacterial diversity. Abbreviations: 5-HT (serotonin), GABA (gamma-aminobutyric acid), ENS (Enteric neuron system), BBB (Brain blood barrier), NLRP3 (NLR family pyrin domain containing 3), IL (interleukin).

**Table 1 ijms-25-05377-t001:** Impact of dietary metabolites on neuroinflammation.

Dietary Metabolites	Effects on CNS	Type of Study	References
SCFAs	Regulation of microglia homeostasis	Mouse model, experimental	[32]
Tryptophan metabolites (indole-3-aldehyde and indole-3-propionic acid)	Binding to astrocyte Ahr Inflammation through TGFα and VEGF-B	Mouse model, experimental	[33]
TMAO Secondary bile acids	Inflammation and metabolic changes through FXR, TGR5, and glucocorticoid receptor	Mouse and human study, experimental	[34]
Ketone bodies (βHB)	Suppress activation of the NLRP3 inflammasomeReduction of IL-1β	Mouse model and human monocytes, experimental	[38,39,40]

Abbreviations: CNS, central nervous system; SCFAs, short-chain fatty acids; Ahr, aryl hydrocarbon receptors; TGFα, transforming growth factor α; VEGF-B, vascular endothelial growth factor B; TMAO, trimethylamine-N-oxide; FXR, Farsenoid X; TGR5, Takeda G-protein-coupled receptor 5; βHB, β-hydroxybutyric acid; IL-1β, interleukin-1β.

**Table 2 ijms-25-05377-t002:** Causes and mechanisms of neuroinflammation.

Causes of Neuroinflammation	Molecular Mechanisms of Neuroinflammation	References
Neurodegenerative diseases Cerebrovascular diseases Mental disorders Headache disorders Traumatic brain injury Peripheral inflammatory diseases Aging High-fat diet Toxicants	Increased brain concentration of pro-inflammatory molecules	[10,46]
	Inflammasome activation	[46]
	Functional activation of microglia and astrocytes	[18,26]
	Lower mitochondrial oxidative capacity	[43]
	Reduced production of SPMs	[10]
	“Leaky” BBB	[20]
	Dysfunction of glymphatic clearance	[23,24]
	Infiltration of peripheral immune cells	[19]
	Protein aggregation	[10,48]
	Glial cell production of neurotoxic factors	[22]
	Reduced expression of BDNF	[43]
	Neuronal cell death	[48]

Abbreviations: SPMs, specialized pro-resolving lipid mediators; BBB, blood–brain barrier; BDNF, brain-derived neurotrophic factor.

**Table 3 ijms-25-05377-t003:** Psychobiotics implicated in the production of neurotransmitters.

Neurotransmitters	Psychobiotics	References
Serotonin (5-HT)	*Streptococcus* spp., *Enterococcus* spp., *Escherichia* spp., *Lactobacillus plantarum*, *Klebsiella pneumoniae*, and *Morganella morganii*	[69]
3,4-Dihydroxyphenethylamine	*Escherichia coli* (*E. coli*), *Proteus vulgaris*, *Serratia marcescens*, *Staphylococcus aureus*, *Hafnia alvei*, and *Klebsiella pneumoniae*	[78]
GABA	*Lactococcus Lactis*, *Lactobacillus Reuteri*, species from the genera bifidobacterium and bacteroides, *Akkermansia Muciniphila*	[82]

## Data Availability

Not applicable.

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
