# Peer review of "Gut–Brain Axis: Focus on Sex Differences in Neuroinflammation"

_ijms, 2024, doi:10.3390/ijms25105377_

Round 1
Reviewer 1 Report
Comments and Suggestions for Authors
Undoubtedly, the concept of this review can be very engaging. However, several crucial points need improvement. First, the review is difficult to follow. It would be greatly appreciated if the authors would focus more on their objectives and avoid repeating the same information multiple times. Additionally, there are too many abbreviations used throughout the manuscript. The first time a contraction appears, the authors should write the full name and then use the abbreviation consistently throughout the rest of the review. This technique will help in avoiding the repetition of both the full name and abbreviation.
- The title should reflect the main essential points referred to in this study. The abstract is too short and should be consistent with this narrative review. The main goal should be the same throughout the manuscript.
- The introduction section should be more informative. The authors should summarise all information covered up to line 229. It would be best to show this data in Tables or Figures. This knowledge is relevant to the extent that it is the previous step for the main goals of this review.
- It would be a good idea if the authors added a subsection describing the methodology used to collect the articles related to their objectives. From what year to what year was this review carried out?
- In the body of this manuscript, the most essential topics that the authors should highlight are: 1. Molecular mechanisms of neuroinflammation, 2. The gut-brain axis, 3. Neuroinflammation and Gender, and 4. Brain-Gut Microbiota Axis and Gender. Since the authors said in their conclusions that "most studies are based on animal models", in each topic presented, it would be a good idea to present the findings in humans first and then in animal models. To better understand where we are in research on these relevant topics.
It would be a good idea if the authors wrote a paragraph of limitations and suggestions before the conclusion. The authors should indicate why this review is crucial and to what extent it contributes to current knowledge of this topic.
I encourage the authors to rewrite this manuscript, thinking about the main objective of this review and responding with the information provided to the most appropriate conclusion of this research work.

Minor editing of English language required.
Author Response
We profoundly thank the reviewers for the comments and useful suggestions aimed at improving the final version of the paper.
This is a point-by-point list of changes made in the paper:
REVIEWER 1
We thank You for your constructive critique and we hope the review process has led to an improved manuscript.
- The title should reflect the main essential points referred to in this study. The abstract is too short and should be consistent with this narrative review. The main goal should be the same throughout the manuscript.
We have expanded the abstract by focusing on the objectives of the paper and modified the title.
- The introduction section should be more informative. The authors should summaries all information covered up to line 229. It would be best to show this data in Tables or Figures. This knowledge is relevant to the extent that it is the previous step for the main goals of this review.
We have expanded the introduction and summarized the information required. We have added a table, as requested.
- It would be a good idea if the authors added a subsection describing the methodology used to collect the articles related to their objectives. From what year to what year was this review carried out?
We have added at the end of the introduction the methodology used to collect the articles.
- In the body of this manuscript, the most essential topics that the authors should highlight are: 1. Molecular mechanisms of neuroinflammation, 2. The gut-brain axis, 3. Neuroinflammation and Gender, and 4. Brain-Gut Microbiota Axis and Gender. Since the authors said in their conclusions that "most studies are based on animal models", in each topic presented, it would be a good idea to present the findings in humans first and then in animal models. To better understand where we are in research on these relevant topics.
We have optimized the division into paragraphs. We have present human findings before animals one. - It would be a good idea if the authors wrote a paragraph of limitations and suggestions before the conclusion. The authors should indicate why this review is crucial and to what extent it contributes to current knowledge of this topic.
We have added a paragraph of limitations and suggestions as requested.
Reviewer 2 Report
Comments and Suggestions for Authors
The review was doggedly done, which might be useful for experts and non-experts. On the other hand, some readers may say that it appeared to be all round or superficial (for example, because the function of ncRNA, miRNA, and IncRNA or apoE4 mutation on neuropathology was not fully detailed in the current form of review).
1. This is based on the narrative review. The authors’ preference could affect the selection of literature. This might be touched on somewhere in the text. If there are positive and negative data, people may select one-side data in the narrative approach.
2. The authors described about the pathophysiological concern of serotonin (i.e., page 8). Headache might be more described.
3. Similarly, depression and anxiety neurosis might be also described.
4. The authors summarized the knowledge in Table 1. The evidence level (strong/weak or much/moderate/mild) in terms of mechanisms could be added even in the authors’ opinions.
5. The onset, initiation, development and progression of gut-brain interplay-associated neuroinflammation could be more detailed; for instance, the whole brain or focal/local brain?
6. Throughout the text, the terms ‘sex and gender’ were mixed; did it mean something in such the mixed use?
7. In many parts throughout the whole text, one paragraph was consisted of one sentence (e.g., row 60-61 in page 2).
8. Row 51-52 in page 2; the first sentence could cite relevant literature.
9. Row 52-54 in page 2; the second sentence could also cite relevant literature.
10. Row 61 in page 2; the sentence could cite relevant literature. The sentence appeared too brief; why is it highlighted?
11. Row 73 in page 2, IL was suddenly described in an abbreviated form. After that, interleukin was abbreviated in page 3.
12. Line 20 in page 12; Transforming Growth Factor could be changed to transforming growth factor in lowercase letters.
13. Page 14; some paragraphs needed the indention (new line) by lowering the beginning the sentence.
14. Line 8 in page 17; the sentence (…IL-1 beta at the end) could cite relevant literature.
15. The title of Table 2 was lacking.
Comments on the Quality of English LanguageMinor editing of English language required; the recheck could be required.
Author Response
We profoundly thank the reviewers for the comments and useful suggestions aimed at improving the final version of the paper.
This is a point-by-point list of changes made in the paper:
REVIEWER 2
We thank You for your constructive critique and we hope the review process has led to an improved manuscript.
- This is based on the narrative review. The authors’ preference could affect the selection of literature. This might be touched on somewhere in the text. If there are positive and negative data, people may select one-side data in the narrative approach.
We have added other references, reporting data present in literature.
- The authors described about the pathophysiological concern of serotonin (i.e., page 8). Headache might be more described.
We have described the serotonergic theory of migraine, with a mention of the role of estrogen.
- Similarly, depression and anxiety neurosis might be also described.
We have added some sentences about the connection between gut brain and the development of anxiety-depressive pathologies.
- The authors summarized the knowledge in Table 1. The evidence level (strong/weak or much/moderate/mild) in terms of mechanisms could be added even in the authors’ opinions.
We have reported literature data in Table 1 (now table2) without adding our opinions.
- The onset, initiation, development and progression of gut-brain interplay-associated neuroinflammation could be more detailed; for instance, the whole brain or focal/local brain?
We have expanded the information on neuroinflammation, as requested. - Throughout the text, the terms ‘sex and gender’ were mixed; did it mean something in such the mixed use?
We have modified the text, as suggested.
- In many parts throughout the whole text, one paragraph was consisted of one sentence (e.g., row 60-61 in page 2).
We have modified the paragraphs, as suggested
- Row 51-52 in page 2; the first sentence could cite relevant literature.
We have added the reference.
- Row 52-54 in page 2; the second sentence could also cite relevant literature.
We have added the reference
- Row 61 in page 2; the sentence could cite relevant literature. The sentence appeared too brief; why is it highlighted?
We have added the reference. The sentence highlighting was eliminated because it was a typo.
- Row 73 in page 2, IL was suddenly described in an abbreviated form. After that, interleukin was abbreviated in page 3.
We have modified, as suggested.
- Line 20 in page 12; Transforming Growth Factor could be changed to transforming growth factor in lowercase letters.
We have modified as suggested
- Page 14; some paragraphs needed the indention (new line) by lowering the beginning the sentence.
We have improved the division of paragraphs as suggested - Line 8 in page 17; the sentence (…IL-1 beta at the end) could cite relevant literature.
We have modified as you suggested
- 15. The title of Table 2 was lacking.
We added the title of table 2 (now table 3).
We thank You for your constructive critique and we hope the review process has led to an improved manuscript.
If additional changes are warranted, we will make them.
We hope that this revised version of our manuscript may now be found suitable for publication.
Sincerely,
Marcello Candelli and Rossella Cianci
Round 2
Reviewer 1 Report
Comments and Suggestions for Authors
Even though the manuscript has improved, the changes are not enough to publish this article: “Gut-brain axis focus on sex-differences in neuroinflammation”. It would be better if the authors consider removing everything that has been highlighted in red/crossed out. The first time a contraction appears, the authors should write the full name.
The intro section is too long and should be rewritten. The information provided in this section should guide readers toward the goal of this review. The authors should focus more on explaining why they believe this topic to be reviewed is crucial and to what extent this information could contribute to the current acknowledgement. It must end with the objective and method: “This narrative review aims to elucidate the intricate interplay between the gut microbiome and the nervous system, describing the molecular pattern underlying neuroinflammation and the role of the brain-gut microbiota axis, from a gender sexual differences perspective. / By using a systematic analysis of relevant literature on the topic of the gut-brain axis, sex -differences, and neuroinflammation, we have conducted a comprehensive search of electronic databases including PubMed, MEDLINE, and Google Scholar. We use keywords for search, such as "neuroinflammation", "sex-differences", "gut microbiome", "gut-brain axis", and "neurotransmitters". We have considered research and review articles, meta-analyses, and systematic reviews written in English over the last fifteen years and in English. We have selected articles based on the relevance of the study design, and methodology, and sample size. All data were analysed to deeper understand the relationship between the gut-brain axis and sex in neuroinflammation for a better- personalized medicine”.
It would be best to highlight the second section as a 'Neuroinflammation' (Line 88-192). This section should be more concise and concrete. The meaning of Tables (1) and figures should be noted in the text. Lines 383-391: this information should be written in this section. Lines 482-491: this information should be written related to line 464. Lines 492-503: this information should be written in the “neuroinflammation and sex-differences”.
In the gut-brain axis section, authors should consider that there are different topics (highlighted in blue) that should be described in subsections. Line 633-634: “A possible explanation is the interaction between intestinal microbiota and microglia.” How? Why?
Line 980-1025: How does this information relate to sex differences in neuroinflammation?
Before the limitations and suggestions paragraphs, the authors should highlight why this review is important and to what extent this information contributes to current knowledge of this topic.
I would suggest authors review what they have written. All information provided must make sense and relate to the main objective of this topic. Basic information should be described in the context of the main objective. Avoid repeating information in the text if it has been shown in figures and tables and written previously.

Minor editing of English language required.
Author Response
Rome, May 6th 2024
Dear Editor of the International Journal of Molecular Science,
First of all, my coauthors and I would like to thank You sincerely for this opportunity, allowing us to resubmit our paper (ID: IJMS-2966595) after extensive revisions, for a possible publication upon IJMS.
We profoundly thank the reviewers for the comments and useful suggestions aimed at improving the paper.
We thank You for your constructive critique which let us improve our manuscript.
We hope this revised version of our manuscript may now be found suitable for publication.
Sincerely,
Rossella Cianci and Marcello Candelli
This is a point-by-point list of changes made in the paper:
REVIEWER 1
Even though the manuscript has improved, the changes are not enough to publish this article: “Gut-brain axis focuses on sex-differences in neuroinflammation”. It would be better if the authors consider removing everything that has been highlighted in red/crossed out. The first time a contraction appears, the authors should write the full name.
Thanks for your suggestions. We have removed all sentences, as requested, except the sentences written following the suggestion of reviewer 2. We have carefully checked all the abbreviations present in the manuscript and modified them as suggested.
The intro section is too long and should be rewritten. The information provided in this section should guide readers toward the goal of this review. The authors should focus more on explaining why they believe this topic to be reviewed is crucial and to what extent this information could contribute to the current acknowledgment. It must end with the objective and method: “This narrative review aims to elucidate the intricate interplay between the gut microbiome and the nervous system, describing the molecular pattern underlying neuroinflammation and the role of the brain-gut microbiota axis, from a gender sexual differences perspective. / By using a systematic analysis of relevant literature on the topic of the gut-brain axis, sex -differences, and neuroinflammation, we have conducted a comprehensive search of electronic databases including PubMed, MEDLINE, and Google Scholar. We use keywords for search, such as "neuroinflammation", "sex-differences", "gut microbiome", "gut-brain axis", and "neurotransmitters". We have considered research and review articles, meta-analyses, and systematic reviews written in English over the last fifteen years and in English. We have selected articles based on the relevance of the study design, and methodology, and sample size. All data were analysed to deeper understand the relationship between the gut-brain axis and sex in neuroinflammation for a better- personalized medicine”.
We have modified the introduction, as suggested.
It would be best to highlight the second section as a 'Neuroinflammation' (Line 88-192). This section should be more concise and concrete. The meaning of Tables (1) and figures should be noted in the text. Lines 383-391: this information should be written in this section. Lines 482-491: this information should be written related to line 464. Lines 492-503: this information should be written in the “neuroinflammation and sex-differences”.
We have modified the second section and named it ‘Neuroinflammation’. We have explained the meaning of Tables and figures.
In the gut-brain axis section, authors should consider that there are different topics (highlighted in blue) that should be described in subsections. Line 633-634: “A possible explanation is the interaction between intestinal microbiota and microglia.” How? Why?
We have created subsections and modified the sentences.
Line 980-1025: How does this information relate to sex differences in neuroinflammation?
Before the limitations and suggestions paragraphs, the authors should highlight why this review is important and to what extent this information contributes to current knowledge of this topic.
I would suggest authors review what they have written. All information provided must make sense and relate to the main objective of this topic. Basic information should be described in the context of the main objective. Avoid repeating information in the text if it has been shown in figures and tables and written previously.
We have made the changes as suggested
Reviewer 2 Report
Comments and Suggestions for Authors
The report has been much improved.
The authors newly created the limitation section. This is a good approach. In this limitation, some problems to select the literature in the narrative review may be moved and described. This is up to the authors. In addition, the authors described the need of research regarding genetics, epigenetic mechanisms, endocrine pathways, and molecular signaling…. The crosstalk between the basic science and clinical studies (e.g., using functional MRI etc.) could be described more. The word “further researches” would be better to modify “further research”.
Comments on the Quality of English LanguageNative check again, please.
Author Response
Rome, May 6th 2024
Dear Editor of the International Journal of Molecular Science,
First of all, my coauthors and I would like to thank You sincerely for this opportunity, allowing us to resubmit our paper (ID: IJMS-2966595) after extensive revisions, for a possible publication upon IJMS.
We profoundly thank the reviewers for the comments and useful suggestions aimed at improving the paper.
We thank You for your constructive critique and hope the review process let us improve our manuscript.
If additional changes are warranted, we will make them.
We hope this revised version of our manuscript may now be found suitable for publication.
Sincerely,
Rossella Cianci and Marcello Candelli
This is a point-by-point list of changes made in the paper:
REVIEWER 2
The authors newly created the limitation section. This is a good approach. In this limitation, some problems to select the literature in the narrative review may be moved and described. This is up to the authors. In addition, the authors described the need of research regarding genetics, epigenetic mechanisms, endocrine pathways, and molecular signaling…. The crosstalk between the basic science and clinical studies (e.g., using functional MRI etc.) could be described more. The word “further researches” would be better to modify “further research”.
We have made the changes as requested.
Round 3
Reviewer 1 Report
Comments and Suggestions for Authors
The manuscript has improved. There are two minor changes that need to be made before publication.
Line 142: T-lymphocytes cells.
Line 159: Table 3 should be placed immediately after referring to it in the text.

Author Response
The manuscript has improved. There are two minor changes that need to be made before publication.
Line 142: T-lymphocytes cells.
Line 19: Table 3 should be placed immediately after referring to it in the text.
Thanks for your suggestions. We have made the changes required.